# Nano-Treating Promoted Natural Aging Al-Zn-Mg-Cu Alloys

Jie Yuan [1,2], Qian Liu [3], Shuaihang Pan [2], Mingjie Xu [4], Narayanan Murali [1,2], Jiaxing Li [3], Shuai Wang [3] and Xiaochun Li [1,2,*]

1   Department of Materials Science and Engineering, University of California Los Angeles, Los Angeles, CA 90095, USA; jie52@g.ucla.edu (J.Y.); narmurali@gmail.com (N.M.)
2   Department of Mechanical and Aerospace Engineering, University of California Los Angeles, Los Angeles, CA 90095, USA; luckypsh@ucla.edu
3   Department of Mechanical and Energy Engineering, Southern University of Science and Technology, Shenzhen 518055, China; 11930548@mail.sustech.edu.cn (Q.L.); lijx3@mail.sustech.edu.cn (J.L.); wangs@sustech.edu.cn (S.W.)
4   Materials Research Institute, University of California Irvine, Irvine, CA 92697, USA; mingjix1@uci.edu
*   Correspondence: xcli@seas.ucla.edu

**Abstract:** Natural aging reduces the cost of alloy manufacturing while saving input energy but takes too long to complete for most Al-Zn-Mg-Cu alloys. Research has proved that nano-treating can facilitate precipitation in heat-treatable alloys. In this study, nano-treated Al-6.0Zn-2.6Mg-xCu samples containing different Cu contents were fabricated to investigate the influence of nano-treating on natural aging. TiC nanoparticles were used for nano-treating. Three cooling conditions after solution treatment (water quenching, air cooling, and as-cast) were investigated to check their quench sensitivities. The study shows the alloy's microstructure was modified by nano-treating, and the growth of dendritic arms was inhibited. Compared to the control samples, nano-treating also increased both the microhardness and tensile strength of the alloy after natural aging. Out of the three different solution treatments, the air-cooled samples presented the highest UTS and microhardness values. The precipitation process was sped up by nano-treating by approximately 50%, and a higher volume fraction of GPII zones were formed in the nano-treated samples. HRTEM results also confirm the formation of more GPI and GPII zones in a nano-treated samples. With the help of natural aging, the Al-6.0Zn-2.6Mg-0.5Cu alloy reached a UTS of 455.7 ± 40.2 MPa and elongation of 4.52 ± 1.34% which makes it a great candidate for a naturally aged Al-Zn-Mg-Cu alloy.

**Keywords:** nature aging; Al-Zn-Mg-Cu alloy; nano-treating; TiC nanoparticles

## 1. Introduction

Various aluminum alloys have been used on air and spacecraft due to their high specific strengths. Many of these components are colossal parts that require heat treatment in large furnaces and quenching tanks, thus significantly increasing their manufacturing costs. Natural aging is an aging process that occurs at room temperature without the need for heat-treating furnaces, thus saving energy and reducing manufacturing costs [1]. Researchers have demonstrated that some alloys, after natural aging, can achieve comparable properties to alloys that have undergone T6 heat treatment [2]. Despite its benefits, the natural aging of many alloys is not often used in industrial applications because it requires too much time [2]. Additionally, for some processes like arc welding, post-weld heat treatment is not favorable. For example, it is difficult to conduct heat treatments after field welding outside the factory. A study on welded Al-4.4Zn-2.0Mg-0.54Mn-0.2Zr alloy showed that its tensile strength continuously increased for 10 years during natural aging [3]. The welding zone achieved a higher strength compared to the base material after post-welding heat treatment (PWHT) [4]. However, PWHT, especially with solution treatment, is hard to accomplish under different constraints [5], such as when field welding is used, when complex weld geometries are involved, when large parts are joined, or when

symmetrical heating is difficult. Therefore, natural aging is also preferred in conjunction with welding processes.

Nano-treating is a process used to modify metals and alloys by adding a small number of ceramic nanoparticles into them [6]. It has been demonstrated that artificial aging can be facilitated by ceramic nanoparticles: Ma, et al. discovered that in an Al-Zn-Mg-Cu alloy reinforced by $TiB_2$ nanoparticles, precipitation was facilitated around $TiB_2$ nanoparticles [7]. Dislocations were generated close to the nanoparticles and became heterogeneous nucleation sites for precipitates. Li, et al. added TiN nanoparticles to an Al-Zn-Mg-Cu alloy and found that nanoparticles promote the uniform precipitation of GP zones [8]. The activation energy for GP-zone formation was reduced by nanoparticles, thus promoting precipitation. Furthermore, in an Al-Cu alloy, smaller and denser θ' precipitates were achieved by adding TiC nanoparticles [9]. Because of their positive effects on artificial aging, it is expected that nanoparticles could be beneficial to natural aging as well.

In this work, 7000-series aluminum alloys, particularly Al-Zn-Mg-(Cu), were nano-treated with TiC to study the effects of nano-treating on natural aging. The literature shows that Cu has a strong interaction with vacancies and increases the quench sensitivity of the alloy [10]. Since a high quench sensitivity necessitates solution treatment, the quench sensitivity of these alloys with different Cu contents was also studied.

## 2. Materials and Methods

### 2.1. Fabrication of Nano-Treated Alloy

An aluminum-matrix nanocomposite containing 3.5 vol.% TiC nanoparticles was supplied by MetaLi LLC, Los Angeles, CA, USA. Pure Mg (>99.95%), Cu (>99.98%), Zn (>99.8%), and Al (>99.8%) were used to fabricate the alloys for study. Pure aluminum was melted at 820 °C in a graphite crucible, and designated amounts of Mg, Zn, Cu, and the Al nanocomposite were then added. The melt was stirred and degassed for 30 mins before casting into a rod-steel mold (Ø50 × 400 mm). Control samples without nanoparticles were fabricated at 750 °C using the same method. A higher processing temperature was applied for nano-treated samples to avoid reactions with TiC in the aluminum melt [11]. Casting experiments were conducted with one thermocouple placed in the center of the mold to measure the cooling rate during solidification. The average cooling rate during solidification was measured at about 2.73 K/s.

### 2.2. Composition and Microstructure

Samples were cut from the center of the ingots for characterization. The chemical composition of the samples was determined by optical emission spectrometry (Spark CCD 7000, NCS), and they are listed in Table 1. Briefly, the samples designated as "-NP" signify nano-treatment with 1 vol.% of TiC nanoparticles. The "A-0Cu" alloys contain 0 wt.% of Cu, the "B-0.5Cu" alloys contain 0.5 wt.% of Cu, and the "C-1Cu" alloys contain 1.0 wt.% of Cu.

**Table 1.** Sample names and composition (unit. wt.%).

| Samples | Al | Zn | Mg | Cu | Ti | Si | Fe | TiC vol.% |
|---|---|---|---|---|---|---|---|---|
| A-0Cu | Bal. | 6.01 ± 0.16 | 2.48 ± 0.07 | 0.00 ± 0.00 | 0.00 ± 0.00 | 0.04 ± 0.01 | 0.03 ± 0.01 | 0.00 ± 0.00 |
| A-0Cu-NP | Bal. | 5.97 ± 0.22 | 2.61 ± 0.10 | 0.00 ± 0.00 | 1.17 ± 0.12 | 0.08 ± 0.01 | 0.10 ± 0.01 | 0.86 ± 0.01 |
| B-0.5Cu | Bal. | 6.07 ± 0.16 | 2.53 ± 0.03 | 0.52 ± 0.02 | 0.00 ± 0.00 | 0.02 ± 0.00 | 0.04 ± 0.00 | 0.00 ± 0.00 |
| B-0.5Cu-NP | Bal. | 5.93 ± 0.23 | 2.74 ± 0.09 | 0.59 ± 0.01 | 1.24 ± 0.04 | 0.06 ± 0.00 | 0.09 ± 0.01 | 0.91 ± 0.00 |
| C-1Cu | Bal. | 6.18 ± 0.15 | 2.63 ± 0.09 | 1.11 ± 0.06 | 0.00 ± 0.00 | 0.03 ± 0.00 | 0.03 ± 0.00 | 0.00 ± 0.00 |
| B-1Cu-NP | Bal. | 6.01 ± 0.29 | 2.49 ± 0.10 | 1.10 ± 0.05 | 1.26 ± 0.09 | 0.07 ± 0.01 | 0.11 ± 0.01 | 0.93 ± 0.01 |

To test quench sensitivity, the different samples were solutionized at 473 °C for 1.5 h and quenched in water (about 40.0 K/s) or air-cooled (2.21 K/s). A thermocouple was attached to the sample to measure the average cooling rate. The samples were ground,

polished, and etched for optical microscopy and scanning electron microscopy (SEM, ZEISS, Supra 40VP).

All samples were naturally aged for 44 days, during which precipitation behavior was observed by differential scanning calorimetry (DSC, TA Instruments, Q600). Samples naturally aged for different durations were heated from 100 °C to 300 °C at a heating rate of 10 K/min, and the latent heat was recorded. The weights of the tested samples were 12–15 mg, and the heat flow was normalized. High-resolution transmission electron microscopy (HRTEM, JEOL, JEM 2800) was also utilized to examine the precipitates. HRTEM samples were first ground to a thin film with a thickness under 100 μm and electropolished by a twin-jet unit (Struers TenuPol) with an electrolyte of 20% nitric acid in methanol at −20 °C.

### 2.3. Mechanical Properties

Mechanical properties of different samples were tested: as-cast samples, water quenched samples after solution treatment and air-cooled samples after solution treatment. As these samples were naturally aging for 44 days, microhardness was tested by a microhardness tester (LECO, LM 800AT) with a load of 100 gf and a dwell time of 15 s. Tensile bars were cut according to ASTM E8 standards by wire electrical discharge machining (EDM). Tensile testing was performed by a tester (WDW-200E, LICHI) at a strain rate of 0.01 mm/mm/min.

## 3. Results and Discussion

### 3.1. Microstructure

Optical images of the fabricated samples are shown in Figure 1 with measured average grain sizes. The black dots in the optical images in Figure 1a–c are eutectic phases with nanoparticles at the grain boundaries. They have small and equiaxed grains, typically between 30–40 μm in size, due to the grain refinement effects of the nanoparticles; the difference in Cu composition did not influence the grain size much. The control samples in Figure 1e–g have large grain sizes with dendritic arms. Adding extra Cu results in thinner dendritic arms and smaller grain sizes, as shown in Figure 1h. Generally, the grain size of the control samples is much larger than that of the nano-treated samples.

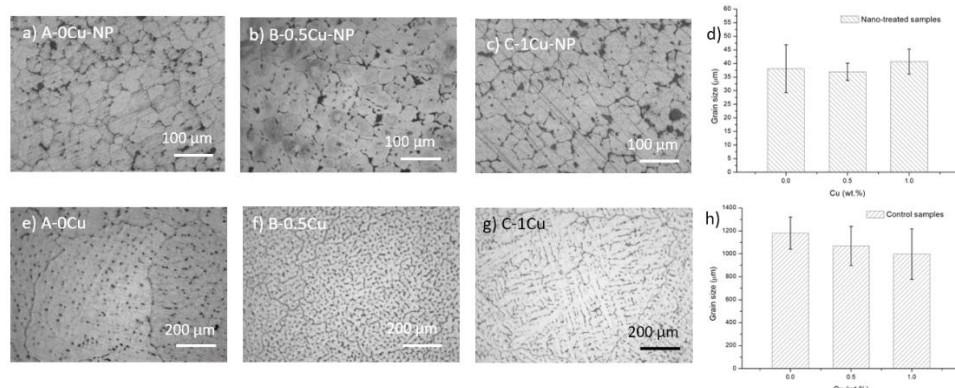

**Figure 1.** Optical images of etched nano-treated samples, (**a**) A-0Cu-NP, (**b**) B-0.5Cu-NP, (**c**) C-1Cu-NP, and (**d**) grain sizes. Optical images of etched control samples, (**e**) A-0Cu, (**f**) B-0.5Cu, (**g**) C-1Cu, and (**h**) their grain sizes.

SEM images of the as-cast and solutionized samples are shown in Figure 2. More details are revealed by the SEM images: in the as-cast samples, the secondary phases in the nano-treated samples are located at grain boundaries. In contrast, most of the secondary phases in the control samples formed isolated islands where liquid pockets existed between dendritic arms during the last stage of solidification. A higher Cu content results in a higher volume fraction of secondary phases after casting, as shown in Figure 2a,e,i and Figure 2c,g,k. After solution treatment at 473 °C for 1.5 h, most secondary phases

were dissolved. Nanoparticle pseudo-clusters are shown in Figure 2b,f,g, and some $Al_3Ti$ intermetallic phases are located inside the grains. There is little difference among the three nano-treated samples after solution treatment. While no residual secondary phase existed in the control samples of A-0Cu and B-0.5Cu after solutionization, there are still a few secondary phases left in the C-1Cu sample. An SEM image with higher magnification is presented in Figure 3a to show the nanoparticle pseudo-cluster after solution treatment. The measured size distribution of the particles is given in Figure 3b, with the average size of these particles being 110.5 $\pm$ 79.0 nm.

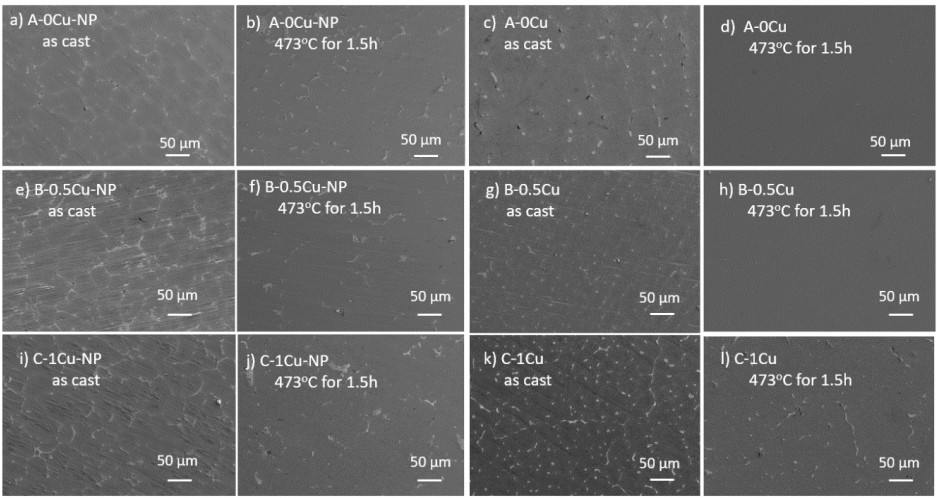

**Figure 2.** SEM images of as-cast samples: (**a**) A-0Cu-NP, (**c**) A-0Cu, (**e**) B-0.5-NP, (**g**) B-0.5Cu, (**i**) C-1Cu-NP, (**k**) C-1Cu. Corresponding SEM images of samples after solution treatment at 473 °C for 1.5 h: (**b**) A-0Cu-NP, (**d**) A-0Cu, (**f**) B-0.5-NP, (**h**) B-0.5Cu, (**j**) C-1Cu-NP, (**l**) C-1Cu.

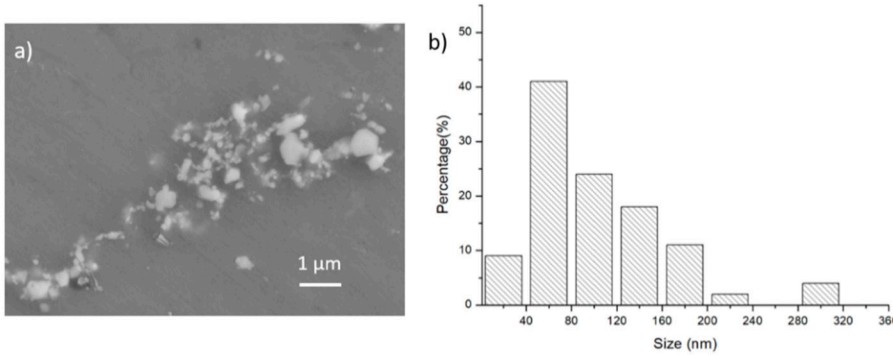

**Figure 3.** (**a**) SEM images of remaining TiC nanoparticles after solution treatment from sample A-0Cu-NP; (**b**) nanoparticle size distribution.

### 3.2. Natural Aging and Mechanical Properties

Samples with different Cu contents were solutionized and either water quenched or air-cooled. Together with the as-cast samples, they were kept at room temperature (25 $\pm$ 5 °C) for 44 days. Microhardness was recorded during the natural aging process, and the hardness evolution curves are presented in Figure 4. There are two peaks accompanied by two valleys in each sample. Peak I indicates the clustering of Zn and Mg elements, and the formation of GPI zones causes Peak II. Valley I is caused by the transition of some clusters to GPI zones, while Valley II is caused by the evolution of some GPI zones to GPII zones. Figure 4a represents the curves of the nano-treated B-0.5Cu-NP sample after different processing conditions. Before natural aging (0 days), the B-0.5Cu-NP sample, after water quenching, shows the lowest microhardness because all secondary phases were dissolved. The sample after air cooling has the highest microhardness because precipitation

happened during cooling. The precipitation speeds within the B-0.5Cu-NP samples in Figure 4a are similar since the positions of the two peaks and two valleys are similar. After 44 days of natural aging, the sample after water quenching and air cooling shows higher microhardness relative to the cast sample. Similar trends were observed in the B-0.5Cu sample, as shown in Figure 4b. Still, Valley I in this sample is not as sharp as Valley I in Figure 4a because the formation of solute clusters and transformation of clusters to GPI zones are happening simultaneously. Valley II occurs after 30 days of natural aging, which is later than Valley II (20 days) in the nano-treated sample, indicating slower precipitation within the control samples. In Figure 4c,d, the cast samples containing different Cu wt.% are compared. The positions of the peaks in the nano-treated samples are similar and, for the most part, occur earlier than the same peaks and valleys in the corresponding control samples, indicating facilitated precipitation by nano-treating. Normally, nano-treated samples exhibit higher microhardness over control samples. However, the microhardness of the Peak I in nano-treated samples was similar or slightly lower than the control ones. This is because the transition from clustering to GPI zones and the clustering of Zn and Mg happened at the same time. Since the transition is promoted in the nano—treated samples, the microhardness of Peak I was subsequently lowered.

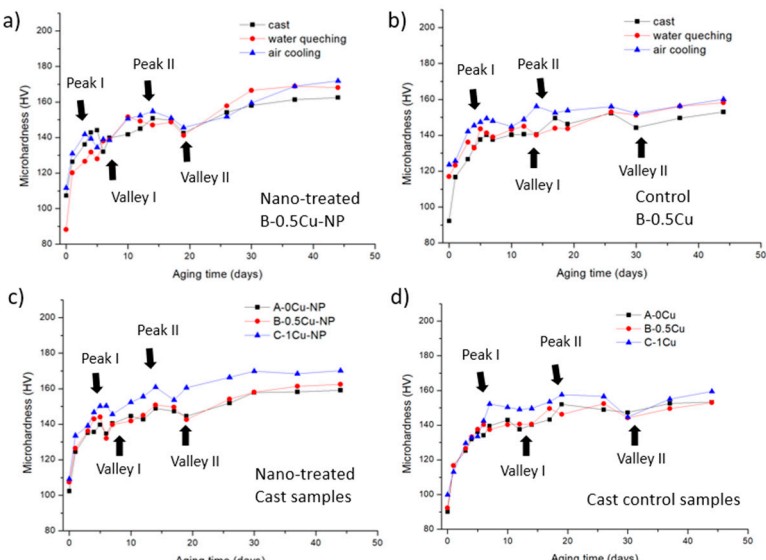

**Figure 4.** (**a**) B-0.5Cu-NP and (**b**) B-0.5Cu sample's hardness during natural aging after casting, water quenching, and air cooling; (**c**) nano-treated samples and (**d**) control samples' hardness during natural aging after cast. Peak I indicates the clustering of solute at-oms; Peak II indicates the formation of GPI; Valley I indicates the transition from clustering to GPI zones; Valley II indicates the transition from GPI zones to GPII zones.

Peak hardness values after 44 days of natural aging are summarized in Figure 5. The nano-treated samples showed higher microhardness values over the control samples because nanoparticles promoted precipitation and refined their microstructures. Additionally, microhardness generally increased with the Cu content. The air-cooled sample offers the highest microhardness in the nano-treated samples, possibly due to extra precipitation during cooling. The cast sample has the lowest microhardness since the secondary phases are not fully dissolved, and thus, fewer solutes precipitate from the matrix. The control samples show a similar trend, but the difference between the air-cooled and water-quenched samples is small. When the copper content falls below 1%, the quench sensitivity is low [12], so the two don't show a great difference. However, the larger difference in the nano-treated samples shows that nanoparticles will increase the quench sensitivity of the alloy.

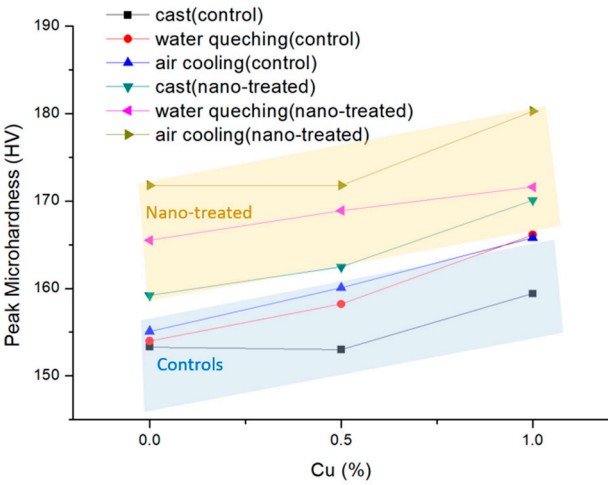

**Figure 5.** Peak microhardness of all samples during natural aging in 44 days.

Tensile testing results are shown in Figures 6–8. The stress-strain curves of B-0.5Cu-NP samples naturally aged for 40 days are shown in Figure 6a, and a comparison of the properties are in Figure 6b. Similar to the microhardness results, the sample after air cooling has the highest ultimate tensile strength (UTS). Due to undissolved secondary phases, the cast sample has the lowest UTS, YS (yield strength), and EL (elongation). The property changes in the B-0.5Cu-NP sample during aging are presented in Figure 7. A longer natural aging time results in better UTS and YS, while the elongation is similar. The strength improvement from 20 to 30 days is larger than the improvement from 30 to 40 days, indicating that the strengthening from GPI zones is better than that from solute clusters, but that the strengthening from GPI zones and GPII zones are similar. The properties of the samples with different Cu weight percentages are shown in Figure 8. Nano-treated samples with higher Cu wt.% have higher UTS, YS, and slightly lower EL values. All the control samples broke before yield due to the existence of casting defects. The yield strengths of these samples are therefore not measured.

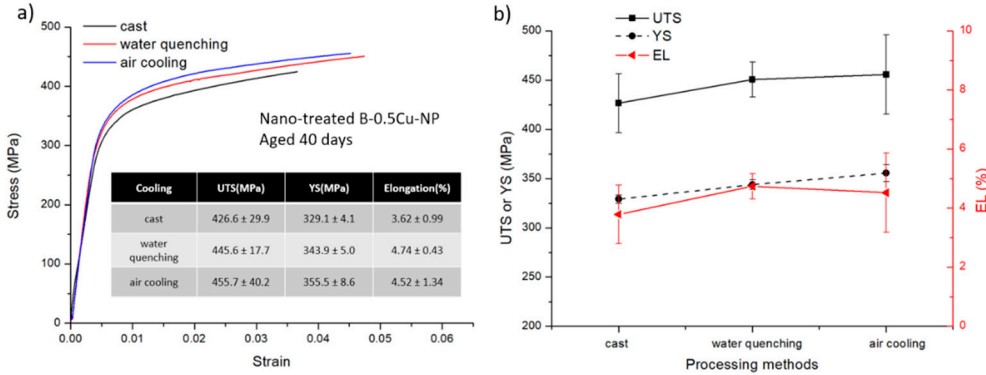

**Figure 6.** (**a**) Tensile testing of the nano-treated B-0.5Cu-NP samples aging for 40 days after casting, water quenching, and air cooling. The inserted table contains tensile data. (**b**) Comparison of the corresponding UTS, YS, and EL values of each sample.

## 3.3. Precipitation Evolution

Precipitation behavior was studied by DSC analysis. B-0.5Cu-NP and B-0.5Cu cast samples naturally aged for different durations were heated from 100 °C to 300 °C at a heating rate of 10 K/min. The recorded heat flow curves are presented in Figure 9, and two peaks are marked in the samples. The first peak, from 120 °C to 160 °C, indicates the dissolution of GPI zones, and a smaller peak, from 180 °C to 210 °C, represents the dissolution of GPII zones [13]. The peak area represents the volume of the corresponding

GP zone. Because these peaks have overlapped, a tangent line connecting both ends of Peak II was drawn, and the area between this tangent line and the peak is measured, as shown in Figure 9a. This is not the absolute value of the GP II volume, but it effectively indicates the difference between samples. Peak II areas are presented in the inserted tables, and they increase with aging duration in both samples, indicating the formation of GPII zones during natural aging. Comparatively, the B-0.5Cu-NP sample has a much larger Peak II area relative to the B-0.5Cu sample. This means the aging speed is facilitated by nano-treating, which is in accordance with the microhardness data.

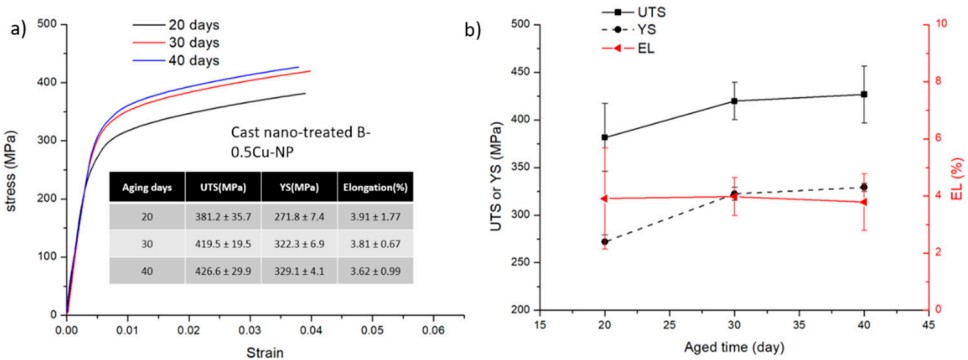

**Figure 7.** (**a**) Tensile testing of the nano-treated B-0.5Cu-NP samples aged 20, 30, and 40 days after casting. The inserted table contains tensile data. (**b**) Comparison of the corresponding UTS, YS, and EL values of each sample.

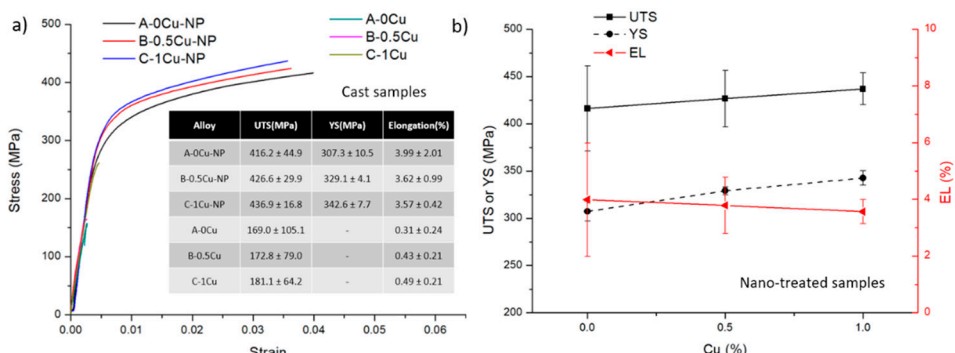

**Figure 8.** (**a**) Tensile testing of all samples aged for 40 days after casting. The inserted table contains tensile data. (**b**) Comparison of the corresponding UTS, YS, and EL values of the nano-treated samples.

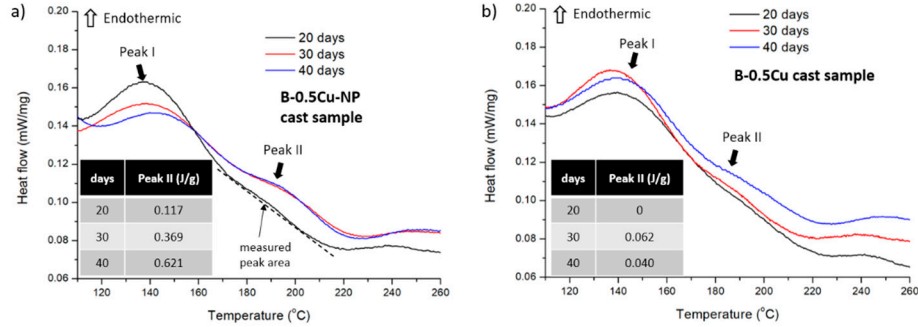

**Figure 9.** Heat flow during heating up the (**a**) B-0.5Cu-NP cast sample and (**b**) B-0.5Cu cast sample with different natural aging days. Peak I is an endothermic peak representing GPI zone dissolution. Peak II is an endothermic peak representing GPII zone dissolution. The inserted table is the calculated peak II area indicating the volume of GPII zones.

The formation of GPII zones was further confirmed by the HRTEM images shown in Figure 10, where the brighter spots are Zn-rich atomic columns [2,14]. The <110>Al direction was chosen for a better GPII observation. According to previous studies, GPI zones are bulky, spherical nanostructures, and GPII zones are elongated nanostructures with a length-to-thickness ratio of over 1.6 with long axes parallel to {111}Al [2,15]. In Figure 10, HRTEM images were taken of the as-cast B-0.5Cu-NP sample natural aged for 40 days, and the bright spots are either GPI or GPII zones uniformly distributed in the matrix. One typical GPII zone is marked in Figure 10b with a size of 1.6 × 4.3 nm.

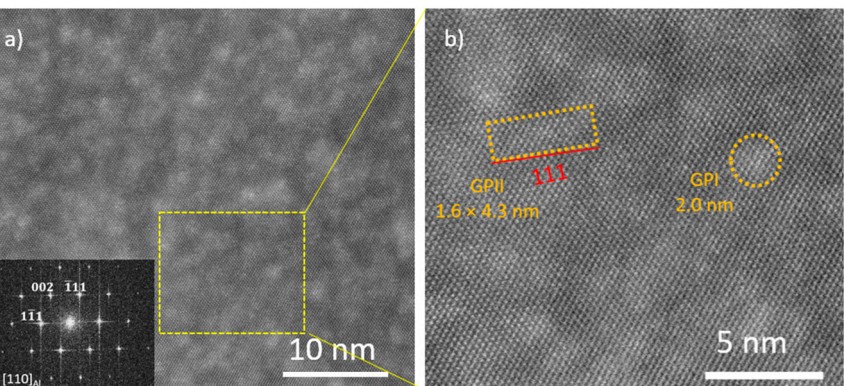

**Figure 10.** (**a**) HRTEM image of the as-cast B-0.5Cu-NP sample natural aged for 40 days. Inserted is the corresponding fast Fourier transform (FFT) pattern. (**b**) Image of the region in the yellow square with marked GPI and GPII zones.

## 4. Conclusions

Nano-treated Al-6.0Zn-2.6Mg-xCu samples containing different Cu contents have been fabricated and studied. The nano-treated samples have a refined microstructure, and the conventional growth of dendritic arms was inhibited. Three cooling conditions (water quenching, air cooling, and as-cast) were selected for natural aging investigations. Compared to the control samples, nano-treating increased both the microhardness and tensile strength of the alloys after natural aging. Air-cooled samples showed the highest UTS, yield, and elongation after 40 days of natural aging; a possible reason is that the precipitation process started during air cooling. Nanoparticles also increased the quench sensitivity such that the microhardness difference between air-cooled samples and water-quenched samples were larger than the differences between their respective control groups. Microhardness evolution curves indicate an earlier formation of GPI zones and GPII zones in the nano-treated samples, occurring in approximately half the time relative to samples without nanoparticles. DSC analysis validated a higher volume of GPII zones that had formed in the B-0.5Cu-NP cast samples compared to the B-0.5Cu cast sample. HRTEM results confirmed the formation of GPI and GPII zones in the B-0.5Cu-NP sample after 40 days of natural aging.

**Author Contributions:** Conceptualization, J.Y.; methodology, J.Y., Q.L., S.P., M.X. and J.L.; validation, J.Y.; formal analysis, X.L.; investigation, J.Y.; resources S.W.; data curation, J.Y.; writing—original draft preparation, J.Y.; writing—review and editing, X.L. and N.M.; visualization, J.Y.; supervision, X.L.; project administration, X.L.; funding acquisition, X.L. All authors have read and agreed to the published version of the manuscript.

**Funding:** This work was partially supported by MetaLi LLC. National Science Foundation, grant number DMR-2011967.

**Acknowledgments:** All authors gratefully acknowledge the assistance of Yang Qiu and Dongsheng He at SUStech Core Research Facilities.

**Conflicts of Interest:** The authors declare no conflict of interest.

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
