# Peer review of "Nano-Treating Promoted Natural Aging Al-Zn-Mg-Cu Alloys"

_jcs, doi:10.3390/jcs6040114_

Round 1
Reviewer 1 Report
This is an interesting and informative article and the manuscript was written in decent English. Authors reported on Nano-Treating Promoted Natural Aging Al-Zn-Mg-Cu Alloys, which fits the scope of Composite Science very well. It can be accepted after minor revision.
- How do you confirm Al3Ti IMC formation in Fig. 2 with EDS or XRD analysis?
- Author should add some statement why microhardness peak 1 in Fig.4b,d is slightly higher than nano-trated samples in Fig.4a,c?
- UTS variations looks very high in Fig. 8 such as 105, 79, 64 MPa. You would better to check again and confirm these variations. Such large changes may be not match with your discussion on the tensile behavior of materials.
- Can you confirm that Fig. 10 is FFT image or inverse FFT?
Author Response
(1)How do you confirm Al3Ti IMC formation in Fig. 2 with EDS or XRD analysis?
Response: Thank you for your question. Al3Ti IMC exists in the aluminum-matrix nanocomposite containing 3.5 vol.% TiC nanoparticles. This nanocomposite was used for alloy fabrication and Al3Ti remained after alloy preparation. Below is the XRD data of the aluminum nanocomposites. The peaks of Al3Ti are marked. Al3Ti is usually an inevitable by-product during the fabrication and processing of Al-Ti system[1]. However, we did not perform the XRD for the alloys samples. On the other hand, EDS cannot distinguish TiC and Al3Ti in the aluminum matrix because they both show Al (from background or Al3Ti) and Ti signals. One way to distinguish TiC and Al3Ti in the aluminum matrix is by their position and geometry. TiC nanoparticles are smaller round shape particles while Al3Ti is larger and rod-like. TiC nanoparticles are at the grain boundaries as they were pushed by the solidification front during solidification[2] while Al3Ti particles are better heterogeneous nucleation sites for alumumin grain so they always exist inside the grain[3].
(2) Author should add some statement why microhardness peak 1 in Fig.4b,d is slightly higher than nano-trated samples in Fig.4a,c?
Response: Thank you for the advice. Peak I indicates the clustering of Zn and Mg elements, and the formation of GPI zones causes Peak II. The explanation to this phenomenon is that both clustering and formation of GPI zones happens at the same time. Because the transition would lower the hardness and the transition in nano-treated samples were faster, The final hardness of the clustering in the nano-treated sample was lowered. We have add the explanation in the manuscript.
(3) UTS variations looks very high in Fig. 8 such as 105, 79, 64 MPa. You would better to check again and confirm these variations. Such large changes may be not match with your discussion on the tensile behavior of materials.
Response: Thank you for the suggestion. The variations are confirmed. The reason of these large variation is that we still have some oxides, impurities left in the alloys and these inclusions made the samples not uniform. We tried degassing and filtering during casting but not successful.
(4) Can you confirm that Fig. 10 is FFT image or inverse FFT?
Response: Thank you. It is FFT image.

Reviewer 2 Report
Authors submitted a generally properly written manuscript entitled 'Nano-treating promoted natural aging Al-Zn-Mg-Cu Alloys'. Presented studies are valuable and propose low-cost aging of aluminum alloys, resulting in a major increase of mechanical properties. However, some minor mistakes have to be improved:
- The introduction provides sufficient background, however, very few literature positions have been cited. Moreover, the citation is a part of a sentence, thusly the dot should be after it (e.g. alloy [7]. instead of alloy.[7]). Please improve it.
- In the methodology section, the manufacturers and models of LM, SEM and HRTEM have to be added.
- Article is full of typos: e.g. line 109 '0.01 mm/mm/min' or line 218 'from 100oC to 300oC'. Please double check your manuscript carefully and correct those typos.
Generally, the paper presents valuable research and I recommend publishing it in the Journal of Composites Science after minor improvement.
Author Response
Reviewer #2:
(1) The introduction provides sufficient background, however, very few literature positions have been cited. Moreover, the citation is a part of a sentence, thusly the dot should be after it (e.g. alloy [7]. instead of alloy.[7]). Please improve it.
Response: Thank you for your advise. We have added more literature positions and moved the citation before period.
(2) In the methodology section, the manufacturers and models of LM, SEM and HRTEM have to be added.
Response: Thank you. The manufacturers and models were added.
(3) Article is full of typos: e.g. line 109 '0.01 mm/mm/min' or line 218 'from 100oC to 300oC'. Please double check your manuscript carefully and correct those typos.
Response: Thank you. We have revised the paper and corrected many typos.
